# Deep learning for the diagnosis of mesial temporal lobe epilepsy

**Kyoya Sakashita**[ID]**, Yukinori Akiyama, Tsukasa Hirano, Ayaka Sasagawa, Masayasu Arihara**[ID]**, Tomoyoshi Kuribara, Satoko Ochi, Rei Enatsu, Takeshi Mikami, Nobuhiro Mikuni**＊

Department of Neurosurgery, Sapporo Medical University, Sapporo, Japan

＊ mikunin@sapmed.ac.jp

**Data Availability Statement:** All relevant data are within the paper. Trained models made from 96 × 96 pixels images of hippocampus https://doi.org/10.6084/m9.figshare.20279559.v1

## Abstract

### Objective

This study aimed to enable the automatic detection of the hippocampus and diagnose mesial temporal lobe epilepsy (MTLE) with the hippocampus as the epileptogenic area using artificial intelligence (AI). We compared the diagnostic accuracies of AI and neurosurgical physicians for MTLE with the hippocampus as the epileptogenic area.

### Method

In this study, we used an AI program to diagnose MTLE. The image sets were processed using a code written in Python 3.7.4. and analyzed using Open Computer Vision 4.5.1. The deep learning model, which was a fine-tuned VGG16 model, consisted of several layers. The diagnostic accuracies of AI and board-certified neurosurgeons were compared.

### Results

AI detected the hippocampi automatically and diagnosed MTLE with the hippocampus as the epileptogenic area on both T2-weighted imaging (T2WI) and fluid-attenuated inversion recovery (FLAIR) images. The diagnostic accuracies of AI based on T2WI and FLAIR data were 99% and 89%, respectively, and those of neurosurgeons based on T2WI and FLAIR data were 94% and 95%, respectively. The diagnostic accuracy of AI was statistically higher than that of board-certified neurosurgeons based on T2WI data (p = 0.00129).

### Conclusion

The deep learning-based AI program is highly accurate and can diagnose MTLE better than some board-certified neurosurgeons. AI can maintain a certain level of output accuracy and can be a reliable assistant to doctors.

**Funding:** The authors received no specific funding for this work.

**Competing interests:** The authors have declared that no competing interests exist.

## 1. Introduction

Temporal lobe epilepsies (TLEs) are classified into three types based on the region of occurrence: the limbic system, mesial TLE (MTLE); neocortex, lateral TLE; and mixed TLE. MTLE involves parts of the limbic system (such as the amygdala and hippocampus) and is thought to be an independent epileptic syndrome associated with unique epileptic seizures and different treatment outcomes. In 1954, Penfield proposed that MTLE is part of a disease group caused by hippocampal sclerosis (HS) [1]. The epileptogenic region in MTLE is located within the limbic system (the hippocampus, amygdaloid body, uncus, and parahippocampal gyrus) of the temporal lobe, whereas that for lateral TLE is located on the lateral surface of the temporal lobe neocortex.

Patients with MTLE experience some distinct symptoms, such as cacosmia, flashback, uneasiness, a sense of fear, and compound visual and auditory hallucinations, perceived as aura. After aural sensation, symptoms of epileptic seizure, such as staring, arrest of movement, and various types of automatisms (such as fumbling for clothes, cheek-biting, and fidgeting), may occur. Lightheadedness persists for 5 to 10 minutes after attack symptoms. The signs of an attack in lateral TLE are acousma (simple sounds such as ringing, but not words) and dizziness.

The most common cause of TLE is HS. HS is diagnosed based on the presence of unilateral hippocampal atrophy and hyperintensities on the T2-weighted magnetic resonance imaging (MRI) or fluid-attenuated inversion recovery (FLAIR) images. In addition, HS is rarely observed in both hippocampi. Seizure control with antiepileptic drug treatment is possible in 70%–80% of patients; however, 20%–30% of patients have drug-resistant intractable epilepsy [2]. Therefore, approximately 10% of all epileptic patients require surgical treatment.

The surgical treatment of TLE results in an 80% reduction in epileptic seizures; however, surgical treatment is less effective against other types of epilepsy. Surgical methods such as anterior temporal lobectomy (ATL) and selective amygdalohippocampectomy are used for treating TLE. Moreover, the surgical resection of mesial structures (hippocampus and amygdala) was first described in the 1950s [3]. Although removal of the lateral temporal cortex was performed during earlier times, the selective removal of the epileptogenic lesion has only recently become mainstream. In ATL, approximately 35–45 mm of tissue from the tip of the temporal lobe cortex toward the inferior horn of the lateral ventricle is removed, and thereafter, the hippocampus and amygdala are resected. ATL provides a relatively wide surgical field for observing the structures. However, this method has certain limitations. Complications such as memory disturbances and upper-quarter homonymous hemianopsia commonly occur after ATL. Moreover, resection on the dominant side may lead to the impairment of the language center located at the tip of the temporal lobe [3]. Hence, neurosurgeons often hesitate to further resect the lateral regions of the temporal lobe cortex.

Furthermore, diagnosing HS is crucial as the hippocampus is important for cognitive functions such as memory and spatial discrimination [4]. In addition, diagnosis of HS can be difficult, even for neurologists or neurosurgeons [5–7]. Neuropathological conditions of HS are mainly characterized by hippocampal neuronal loss. In 2013, the International League Against Epilepsy proposed a classification system for identification of pathological subtypes of HS based on the patterns of neuronal loss [8]. Atrophy and/or high signal intensity of hippocampi on T2-weighted imaging (T2WI) and FLAIR are the most reliable findings of HS. These findings reflect the neuropathological features of HS [9].

In Japan, diagnosis based on neuroimaging is usually performed by neurosurgeons. The 2018 report of Labor Standards Bureau, Ministry of Health, Labor and Welfare, revealed that the number of radiologists in Japan was 6,813, whereas the number of doctors in Japan was

311,963. Therefore, Japanese doctors should be able to reach a diagnosis using neuroimages by themselves.

In this study, we present an alternative diagnosis method using artificial intelligence (AI), which involves automatic detection of the hippocampus and diagnosis of MTLE with the hippocampus as the epileptogenic area. Here, we demonstrated that the deep learning-based AI program successfully detects the hippocampus and therefore, epileptic attacks because of MTLE with the hippocampus as the epileptogenic area based on MRI, with an accuracy of > 90%.

## 2. Methods

### 2.1. Patients

The human and animal studies were approved by the Ethics Committee of Sapporo Medical University Hospital. The study was conducted in accordance with the ethical principles of the Declaration of Helsinki (1964) and its later amendments. The need for consent was waived by the ethics committee.

This study included all consecutive patients diagnosed with epilepsy at our hospital between January 2016 and December 2019 who had not been previously treated. A total of 259 patients (128 men and 131 women) were enrolled and analyzed retrospectively. Forty-six patients were diagnosed with MTLE, whereas 12 patients were diagnosed with MTLE with the hippocampus as the epileptogenic area (Fig 1). Patients diagnosed with MTLE with the hippocampus as the epileptogenic area were selected because they had hippocampal atrophy, hyperintensities on FLAIR MRI and spikes in the mesial temporal lobe on intracranial electroencephalography (EEG). Patients who were found to be seizure-free in a presurgical evaluation with subdural electrodes underwent resection surgeries at the Sapporo Medical University. Two or three continuous coronal sections of T2WI at the level of the anterior commissure from patients with MTLE and those with other diseases such as psychogenic nonepileptic seizures (control subjects) were prepared for AI analysis. MR images were allocated to two datasets: learning and test datasets. The learning data were used for deep learning, whereas the test data were used for analyzing the diagnostic accuracy (i.e., validation). The test data included 30% of all data. MRI data of patients with MTLE with the hippocampus as the epileptogenic area included four images on T2WI (25.0%) and six images on FLAIR (21.4%). MRI data of control subjects included 77 images on T2WI (36.5%) and 77 images on FLAIR (27.7%). After the deep learning process, the diagnostic accuracy was calculated using the test data from T2-weighted images and FLAIR images.

All data were fully anonymized before we accessed them.

### 2.2. MRI examination

The basis for pre-processing was T2WI of the head using a clinical 1.5- or 3.0-Tesla magnetic resonance scanner (Signa HDxt 3.0 Tversion16®, GE Healthcare, Connecticut, USA). The imaging parameters for the T2-weighted fast-spin echo imaging were as follows: flip angle, 90˚; repetition time, 5000 ms; echo time, 102.0 ms; bandwidth, 50.0 kHz; field of view, 200 mm × 200 mm; scan thickness, 4.0 mm; slice gap, 1.0 mm; number of slices, 26–30; matrix, 352 × 256; number of signals averaged, 1; and imaging time, 1 min 05 s. The imaging parameters of FLAIR imaging were as follows: flip angle, 90˚; time of repetition, 5000 ms; echo time, 102.0 ms; bandwidth, 50.0 kHz; field of view, 200 mm × 200 mm; scan thickness, 4.0 mm; slice gap, 1.0 mm; number of slices, 26–30; matrix, 352 × 256; number of signals averaged, 1, and imaging time = 1 min 05 s.

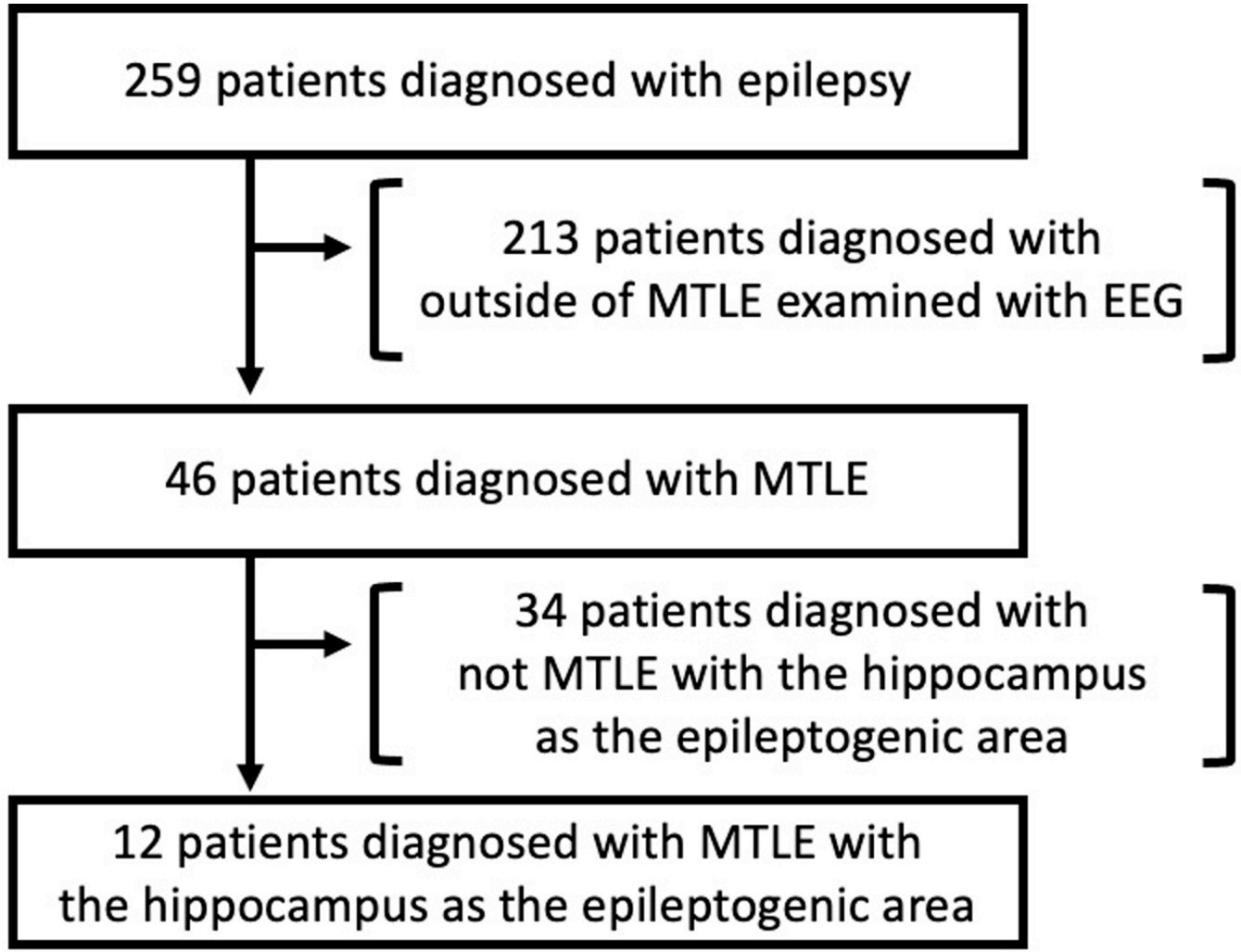

**Fig 1. Flow chart for the diagnosis of MTLE with hippocampal epileptogenic area by artificial intelligence (AI) and board-certified neurosurgeons.** The data of 259 patients with epilepsy were retrospectively analyzed. However, 213 patients without MTLE were excluded. Twelve patients with confirmed diagnosis of MTLE, with the epileptogenic area in the hippocampus, were enrolled in the study.

### 2.3. Data processing

We used an HP Z840 workstation (Hewlett-Packard Company, Palo Alto, California, USA) with a Core Xe-6700 K 4.00 GHz (Intel, Santa Clara, CA, USA) central processing unit, 64 GB of random access memory, and a GeForce GTX 1080 (NVIDIA, Santa Clara, CA, USA) graphics processing unit for the training phase of deep learning. The image sets were processed using a code written in Python 3.7.4 (http://www.python.org) and Pillow 3.3.1 (http://pypi.python.org/pypi/Pillow/3.3.1), which is a python imaging library. We used the Open Computer Vision (OpenCV) 4.5.1 (https://docs.opencv.org/4.5.1/) and Keras (version 2.3.0), a framework for neural networks and a part of the Tensorflow (version 1.14.0) platform. OpenCV is a library of programming functions that is free for use under an open-source BSD license (OpenCV. Open Source Computer Vision Library. 2015.). OpenCV was used for hippocampal detection. MRI images were obtained of 113 T2WI and 148 FLAIR cases with a diagnosis of epilepsy from January 2016 to December 2019. On both the left and right sides, 226 T2WI and 296 FLAIR images with a resolution of 96 × 96 pixels were used for hippocampal

images, and 327 T2WI and 367 FLAIR images with a resolution of 96 × 96 pixels were used for non-hippocampal images after amplification using augmentation. Haar-like features and local binary patterns (LBPs) were used as features. Hyperparameters of the created cascade classifier, mainly scaleFactor, minNeighbors, and minSize, were adjusted for optimization. Image processing was performed separately for the training and test image sets to prevent overlearning during the training phase. Data augmentation was performed for the training data sets; thus, the number of images from patients with MTLE with the hippocampus as the epileptogenic area and from control subjects increased from 12 to 1134 and 134 to 7344 for T2WI, and 22 to 1975 and 201 to 8635 for FLAIR, respectively. In addition, data augmentation was performed for the validation data sets; thus, the number of images from patients with MTLE with the hippocampus as the epileptogenic area and from control subjects increased from 4 to 392 and 77 to 5359 for T2WI, and 6 to 580 and 77 to 5369 for the FLAIR images, respectively. After data augmentation, the total number of images in the training dataset for T2WI was 8478, and for FLAIR was 10610, and that in the validation dataset for T2WI was 5751 and for FLAIR was 5949 (Tables 1 and 2).

The image data generator created varied images using rotation, width and height shift, horizontal and vertical flip, zoom, and shear. This type of data augmentation is commonly used in deep learning (the code is available at https://keras.io/) for small datasets such as those of medical diseases.

The authors selected the VGG16 model because it has been trained on big data and its weights from the pre-training can be used by end-users from the internet to analyze data. The output data were compared with the initial teacher data (two categories–MTLE with the hippocampus as the epileptogenic area and controls). The deep learning model, which was a fine-tuned VGG16 model, comprised several layers (six convolutional layers, three maximum pooling layers, and three fully connected layers). Details of the neural network used in the model are shown in Fig 2. The number of epochs used for each study was 20. We used SGD as an optimizer for the deep learning model.

## 2.4. Visualization of deep learning

Explanation of the output of a deep network remains a challenge. The classification of images with AI is presented in the black box. Convolutional neural networks include multiple layers. An image classifier identifies pixels that have significant influence on the decision making. Therefore, we attempted to visualize the image processing using deep learning neural networks and determine the mechanisms used to classify or diagnose based on an image in the validation dataset. Here, we focused on the last layer to visualize the area of interest used by AI. One such visualization method for deep learning is the VGG16-based gradient-weighted class activation mapping (Grad-CAM) [10]. We used the Grad-CAM method to visualize the areas of interest to distinguish between MTLE and control using the AI program.

**Table 1. Image processing.**

| | MTLE with the hippocampus as the epileptogenic area | | | control | | | total | | |
| --- | --- | --- | --- | --- | --- | --- | --- | --- | --- |
| | training | | test | training | | test | training | | test |
| | original | augmentation | original | original | augmentation | original | original | augmentation | original |
| T2WI | 12 | 1134 | 4 | 134 | 7344 | 77 | 146 | 8478 | 81 |
| FLAIR | 22 | 1975 | 6 | 201 | 8635 | 77 | 223 | 10610 | 83 |

**Table 2. Validation data set.**

| sex | age | laterality | Diagnosis | EEG findings (spikes regional) | symptoms, seizure type |
|-----|-----|-----------|-----------|-------------------------------|------------------------|
| M | 9 | left | MTLE with the hippocampus as the epileptogenic area | left temporal & spike and wave generalized | dialeptic seizure, oral automatism, automotor seizure |
| F | 20 | left | MTLE with the hippocampus as the epileptogenic area | left temporal | abdominal aura, dialeptic seizure, focal impaired awareness seizure |
| F | 22 | left | MTLE with the hippocampus as the epileptogenic area | left temporal | dialeptic seizure, GTCS |
| M | 22 | right | MTLE with the hippocampus as the epileptogenic area | right temporal | dialeptic seizure, automotor seizure |
| F | 22 | right | MTLE with the hippocampus as the epileptogenic area | right temporal | automotor seizure |
| M | 30 | right | MTLE with the hippocampus as the epileptogenic area | right front-temporal | dialeptic seizure |
| M | 38 | right | MTLE with the hippocampus as the epileptogenic area | right front-temporal | dialeptic seizure, GTCS |
| F | 39 | left | MTLE with the hippocampus as the epileptogenic area | left temporal | abdominal aura, aphasic |
| M | 42 | right | MTLE with the hippocampus as the epileptogenic area | right temporal | abdominal aura, dialeptic seizure, automotor seizure |
| F | 45 | right | MTLE with the hippocampus as the epileptogenic area | right and left temporal | abdominal aura, automotor seizure |

MTLE, mesial temporal lobe epilepsy; GTCS, generalized tonic-clonic convulsion; EEG, electroencephalogram.

## 2.5. Diagnosis by six board-certified neurosurgeons

The analysis and diagnosis of MTLE based on T2 and FLAIR were performed by six independent investigators who were board-certified neurosurgeons (Fig 3). HS was diagnosed based on the presence of following characteristics: hippocampal atrophy, hippocampal hyperintensity on T2WI, and other hippocampal signal alterations such as loss of internal architecture of the hippocampus [11,12]. Three epilepsy specialists independently confirmed the diagnosis of MTLE with the hippocampus as the epileptogenic area based on MRI and EEG data.

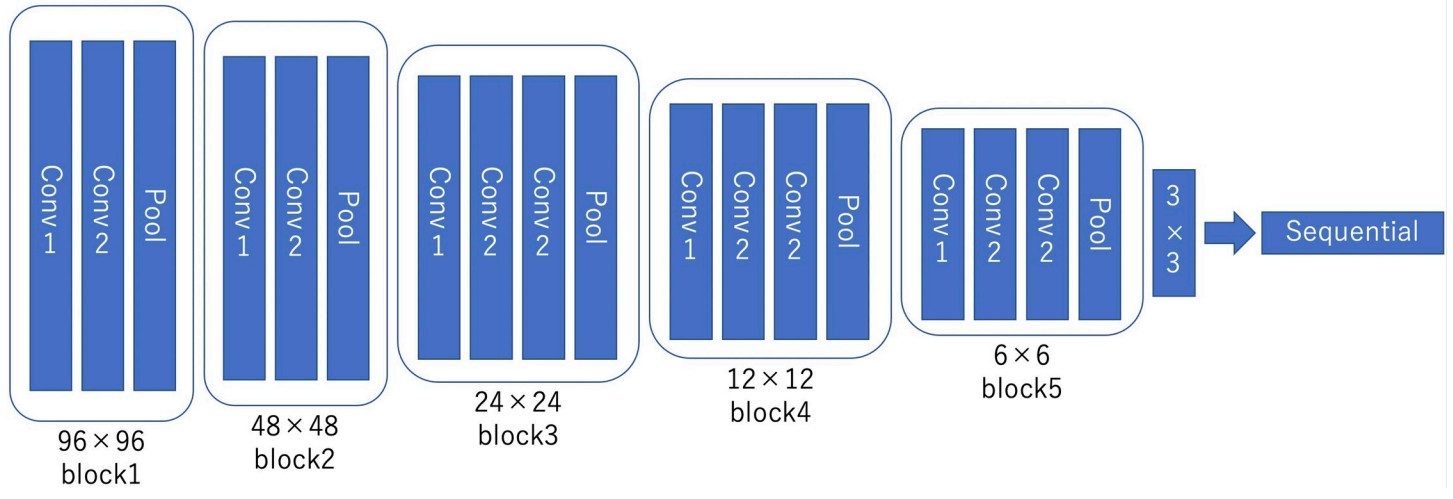

**Fig 2. Scheme of the convolutional neural network.**

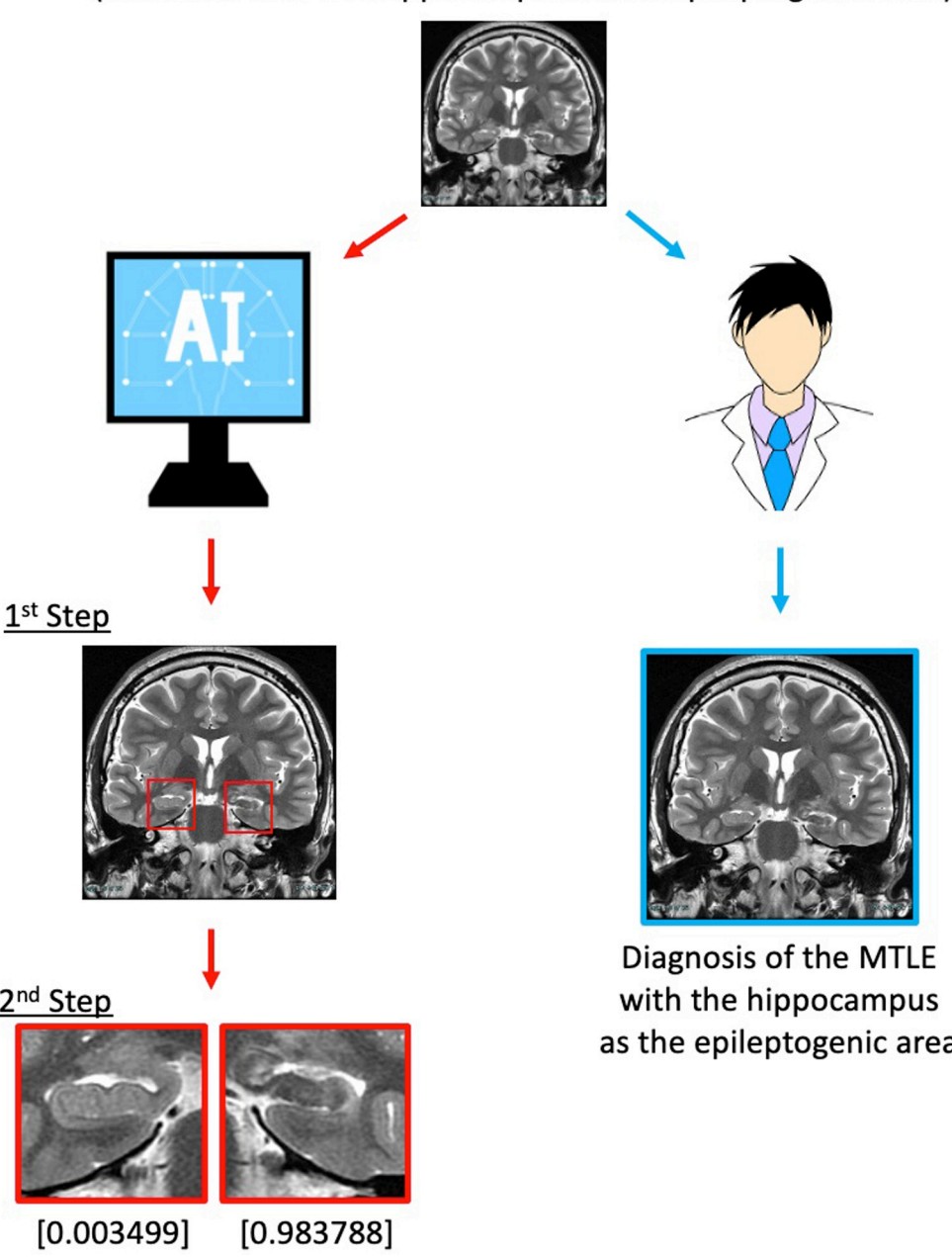

**Fig 3. Scheme of hippocampal detection and prediction of MTLE with the hippocampus as the epileptogenic area.**
In the AI program, the first step is to automatically detect the hippocampus on conventional T2WI and FLAIR coronal
sections. The second step is to diagnose MTLE with the hippocampus as the epileptogenic area and predict the
probability of the disease. Six board-certified neurosurgeons diagnosed MTLE with the hippocampus as the
epileptogenic area using an original image of one slice.

**2.5.1. Selected images.**    We selected 20 MRI images for T2WI and FLAIR (10 images of
patients diagnosed with MTLE with the hippocampus as the epileptogenic area and 10 patients
diagnosed with other epilepsies). These 20 images represented the data of the group (Table 3).

**Table 3. Summary of the findings of the T2-weighted and fluid-attenuated inversion recovery images of the 20 patients.**

| T2 number | FLAIR number | sex | age | laterality | diagnosis | EEG findings (spikes regional) | symptoms, seizure type |
|---|---|---|---|---|---|---|---|
| 1 | 1 | F | 37 | left | MTLE with the hippocampus as the epileptogenic area | left temporal | left conjugate deviation, oral automatism |
| 2 | 2 | F | 39 | left | MTLE with the hippocampus as the epileptogenic area | left temporal | abdominal aura, aphasic |
| 3 | 3 | M | 9 | left | MTLE with the hippocampus as the epileptogenic area | left temporal & spike and wave generalized | dialeptic seizure, oral automatism, automotor seizure |
| 4 | 4 | M | 27 | left | MTLE with the hippocampus as the epileptogenic area | left front-temporal | olfactory aura, dialeptic seizure, GTCS |
| 5 | | F | 22 | left | MTLE with the hippocampus as the epileptogenic area | left temporal | dialeptic seizure, GTCS |
| | 5 | M | 22 | right | MTLE with the hippocampus as the epileptogenic area | right temporal | dialeptic seizure, automotor seizure |
| 6 | | F | 20 | left | MTLE with the hippocampus as the epileptogenic area | left temporal | abdominal aura, dialeptic seizure, focal impaired awareness seizure |
| | 6 | M | 17 | right | MTLE with the hippocampus as the epileptogenic area | right front-temporal | autonomic aura, left versive, dialeptic seizure, left arm tonic seizure |
| 7 | | M | 22 | right | MTLE with the hippocampus as the epileptogenic area | right temporal | dialeptic seizure, automotor seizure |
| 8 | 7 | M | 42 | right | MTLE with the hippocampus as the epileptogenic area | right temporal | abdominal aura, dialeptic seizure, automotor seizure |
| | 8 | F | 45 | right | MTLE with the hippocampus as the epileptogenic area | right and left temporal | abdominal aura, automotor seizure |
| 9 | 9 | M | 30 | right | MTLE with the hippocampus as the epileptogenic area | right front-temporal | dialeptic seizure |
| 10 | | M | 38 | right | MTLE with the hippocampus as the epileptogenic area | right front-temporal | dialeptic seizure, GTCS |
| | 10 | F | 22 | right | MTLE with the hippocampus as the epileptogenic area | right temporal | automotor seizure |
| 11 | | F | 32 | right | EBV encephalitis | right front-temporal | GTCS |
| 12 | | M | 33 | | Lennox-Gastaut syndrome | left temporal & sharp-and-slow-wave generalized | Tonic seizure, infantile spasm |
| 13 | | F | 29 | | Juvenile myoclonic epilepsy | generalized | dialeptic seizure, GTCS |
| 14 | | M | 48 | right | right TLE | right temporal & paroxysmal fast lateralized right | abdominal aura, dialeptic seizure |
| 15 | | F | 30 | | PNES | no epileptic discharge | GTCS |
| 16 | | F | 35 | | right TLE | right temporal | GTCS |
| 17 | | M | 61 | right | meningioma | no epileptic discharge | Tonic seizure, dialectic seizure |
| 18 | | F | 22 | | Juvenile myoclonic epilepsy | spike-and-wave complex &polyspike generalized | dialeptic seizure |
| 19 | | F | 57 | left | left TLE, left amygdala hypertrophy | no epileptic discharge | dialeptic seizure, oral automatism |
| 20 | | F | 7 | right | right OLE | right occipital | blindness |
| | 11 | M | 20 | | meningioma, epilepsy | no epileptic discharge | GTCS |
| | 12 | M | 14 | right | right TLE, cavernoma | spikes regional right front-temporal | dialeptic seizure, automotor seizure |
| | 13 | M | 36 | left | left TLE | sharp wave regional left temporal | unclassified aura, automotor seizure, GTCS |
| | 14 | M | 21 | left | epilepsy | sharp wave lateralized left | bilateral asymmetric tonic, GTCS |
| | 15 | F | 16 | | PNES | no epileptic discharge | dialeptic seizure |
| | 16 | M | 18 | right | right TLE, right amygdala hypertrophy | spikes regional right temporal | dialeptic seizure, GTCS |
| | 17 | M | 67 | left | left TLE, left amygdala hypertrophy | sharp wave regional left temporal | dialeptic seizure, automotor seizure |
| | 18 | M | 25 | | PNES | no epileptic discharge | twitching, stop working |

(*Continued*)

**Table 3.** (Continued)

| T2 number | FLAIR number | sex | age | laterality | diagnosis | EEG findings (spikes regional) | symptoms, seizure type |
|---|---|---|---|---|---|---|---|
| | 19 | F | 5 | right | right FLE | spikes regional right frontal and occipital | bilateral arm tonic seizure, dialeptic seizure |
| | 20 | M | 19 | right | left TLE | repetitive spike regional right fronto-temporal | autonomic aura, dialeptic, GTCS |

20 patientws including 10 MTLE with the hippocampus as the epileptogenic area (case 1–10) and 10 non-MTLE with the hippocampus as the epileptogenic area (case 11–20).

MTLE, mesial temporal lobe epilepsy; GTCS, generalized tonic-clonic convulsion; PNES, psychogenic non-epileptic seizures.

**2.5.2. Patients diagnosed with MTLE.** Retrospectively, 46 patients were diagnosed with MTLE at our institution. However, we excluded cases with no imaging, cases in which surgery was performed, and cases for which no ECoG recording with subdural electrodes existed. Finally, T2WI was validated in 28 cases and FLAIR in 34 cases.

## 2.6. Statistical analysis

Data are presented as mean ± standard deviation or standard error. Statistical analysis was performed using IBM SPSS 22.0 (IBM Corp. Armonk, NY, USA). The Shapiro–Wilk test was used to confirm whether the data followed a normal distribution. When the data distribution was not normal, statistical differences were assessed using the Mann-Whitney U test. We also performed a cross tabulation. A p value of <0.01 was considered statistically significant.

## 3. Results

### 3.1. Stratification of the patients

This study enrolled 12 patients who were diagnosed with MTLE with the hippocampus as the epileptogenic area based on MRI and EEGs by three epilepsy specialists. To assess the diagnostic accuracy, the patients were divided into two groups: those with MTLE with the hippocampus as the epileptogenic area and control subjects. Imaging data of both the groups were further divided into the learning and validation datasets. The validation dataset comprised approximately 30% the data of both the groups (patients with MTLE with the hippocampus as the epileptogenic area and control subjects). The validation dataset included data of randomly selected patients. Patients' data in the validation and learning datasets were statistically matched for age and sex.

Bilateral hippocampi were automatically and successfully detected by the AI program, with an accuracy of 96%–99% using T2WI and of 89% using FLAIR images.

Haar-like feature was selected because it is more accurate than LBP when the hyperparameters are changed to optimize the detection rate and reduce false positives. FLAIR had a 89% detection rate and 0.25 false positives per picture. While checking the detection rate and false positive rate, we examined the effect of changing mainly the following parameters as optimization factors: scaleFactor, minNeighbors, minSize, and maxSize. For the T2WI classifier, scaleFactor was set to 1.06, minNeighbors to 3, minSize to (60,60), and maxSize to (130,130). Similarly, for FLAIR, we set the scaleFactor to 1.1202, minNeighbors to 3, minSize to (60,60), and maxSize to (130,130).

We compared the diagnoses of MTLE with the hippocampus as the epileptogenic area between six board-certified neurosurgeons and AI for each MRI sequence (T2WI and FLAIR). The diagnostic accuracy of AI using T2WI data was 94% and using FLAIR data was

95% (Fig 4). When the diagnostic accuracy was measured using only the original images as the test data, the diagnostic accuracy using T2WI data was 98%, and that using FLAIR data was 95%. However, this result is considered to be insufficiently evaluated because of the small number of original images.

We performed a cross-validation analysis, and the system showed 90–93% accuracy for the T2WI data and 89–95% accuracy for the FLAIR data. For MTLE with the hippocampus as the epileptogenic area, 3 out of 9 T2 cases and 10 FLAIR cases were selected as validation data. Verification was performed five times by replacing the data. Allocation was done using a random number table. The relatively high accuracy from the beginning of the epoch may be due firstly to the fact that VGG16 was a very good model for MTLE differentiation, and secondly to the fact that the image size was small (96 × 96 pixels) and the accuracy was high. However, considering that diagnostic accuracy increased with each successive epoch, we believe that there was a learning effect.

## 3.2. Selected images

The disease (MTLE with the hippocampus as the epileptogenic area) probability in the extracted hippocampus was found to be 92.5% based on T2WI and 82.5% based on FLAIR using the AI program. On the other hand, the disease (MTLE with the hippocampus as the epileptogenic area) probability was 74.1% based on T2WI and 74.2% based on FLAIR according to the six board-certified neurosurgeons. Those judged to have a 50% or greater likelihood of hippocampal sclerosis, according to the AI diagnosis, were treated as correct answers. Those that were not hippocampal sclerosis were treated as correct if the likelihood of hippocampal sclerosis was judged to be less than 50%. Interestingly, diagnosis with AI was good in all, except one, cases; one of the 20 images showed false positives on T2WI and seven of the 20 images on FLAIR. Diagnoses by the board-certified neurosurgeons based on T2 and FLAIR were not different.

The images for which MTLE with the hippocampus as the epileptogenic area was misdiagnosed as normal, or *vice-versa*, always showed a tendency towards hyperintensity on T2WI. Diagnoses by AI were significantly more accurate than those by board-certified neurosurgeons based on T2WI images (p = 0.0001, effect size 0.6152); however, the difference in diagnostic

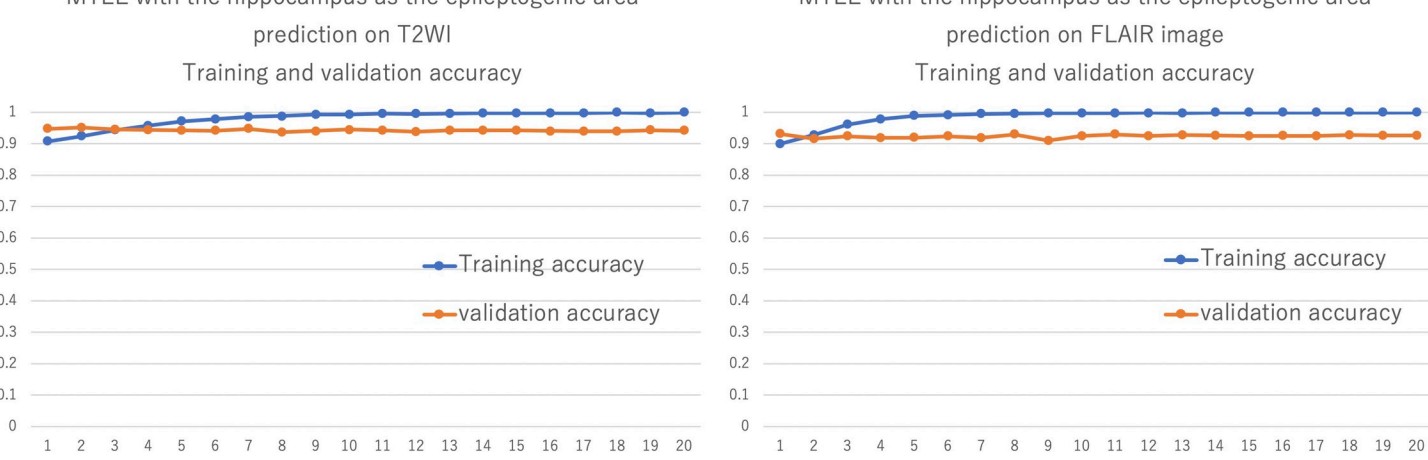

**Fig 4. Validation of overall diagnostic accuracy.** The diagnostic accuracy of the deep learning model for MTLE with the hippocampus as the epileptogenic area based on T2WI was 94.13% (left) and on FLAIR was 94.91% (right).

accuracies of AI and neurosurgeons based on FLAIR was not statistically significant (p = 0.0520, effect size 0.3072) (Table 4).

However, the sensitivity, specificity, and F-value of AI for T2WI were 0.8000, 0.9667, and 0.8421, respectively, and those of physicians for T2WI were 0.5000, 0.5667, and 0.5172, respectively. Similarly, for FLAIR, the sensitivity, specificity, and F-value for the diagnosis by AI were 0.3077, 1.0000, and 0.4706, respectively, whereas those by physicians were 0.5833, 0.5833, and 0.5833, respectively. Although the sensitivity, specificity, and F-value of AI for T2WI were high, the sensitivity for FLAIR was low.

### 3.3. Patients diagnosed with MTLE

The Shapiro–Wilk test showed that the data did not follow a normal distribution (p<0.001); therefore, Mann–Whitney U test was performed. The diagnostic accuracy for MTLE by AI was higher than that by physicians for both T2WI and FLAIR (T2, p = 0.0034; effect size 0.3917, FLAIR, p = 0.0006; effect size, 0.4147). AI diagnosis was superior in the cross tabulation (p<0.001) as well.

However, the sensitivity, specificity, and F-value of AI for T2WI were 0.5556, 0.9362, and 0.5582, respectively, and those of physicians for T2WI were 0.5000, 0.6754, and 0.4576, respectively. Similarly, in FLAIR, the sensitivity, specificity, and F-value for the diagnosis by AI were 0.4, 1.0000, and 0.5714, respectively, whereas those by physicians were 0.6833, 0.6181, and 0.5256, respectively. Although the F-value for AI was high, the sensitivity was low.

**Table 4. Diagnostic accuracy of artificial intelligence & neurosurgeons.**

| case | FLAIR | | | T2WI | | |
|---|---|---|---|---|---|---|
| | AI | Neurosurgeons | (p = 0.0520) | AI | Neurosurgeons | (p = 0.0001) |
| 1 | 0.95 | 0.67 | | 0.99 | 0.50 | |
| 2 | 0.25 | 0.83 | | 1.00 | 0.17 | |
| 3 | 0.90 | 1.00 | | 0.91 | 0.83 | |
| 4 | 0.01 | 0.33 | | 0.98 | 0.83 | |
| 5 | 0.02 | 0.33 | | 0.77 | 0.50 | |
| 6 | 0.00 | 0.83 | | 1.00 | 0.67 | |
| 7 | 0.06 | 0.50 | | 0.99 | 0.17 | |
| 8 | 0.25 | 0.50 | | 0.01 | 0.50 | |
| 9 | 0.07 | 0.17 | | 0.95 | 0.50 | |
| 10 | 0.23 | 0.67 | | 0.04 | 0.33 | |
| 11 | 0.00 | 0.50 | | 0.00 | 0.50 | |
| 12 | 0.00 | 0.83 | | 0.00 | 0.33 | |
| 13 | 0.00 | 0.33 | | 0.00 | 0.83 | |
| 14 | 0.00 | 0.83 | | 0.01 | 0.83 | |
| 15 | 0.00 | 0.83 | | 0.00 | 0.33 | |
| 16 | 0.02 | 0.67 | | 0.00 | 0.83 | |
| 17 | 0.01 | 0.00 | | 0.00 | 0.50 | |
| 18 | 0.00 | 0.33 | | 0.48 | 0.67 | |
| 19 | 0.00 | 0.83 | | 0.01 | 0.33 | |
| 20 | 0.01 | 0.67 | | 0.05 | 0.50 | |

20 patientws including 10 MTLE with the hippocampus as the epileptogenic area (case 1–10) and 10 non-MTLE with the hippocampus as the epileptogenic area patients (case 11–20).

### 3.4. Diagnostic site of MTLE with the hippocampus as the epileptogenic area

AI focused on the color gradient of the area of interest for diagnosing the epileptogenic area in the hippocampus (Fig 5). The focus was expected to be on the outer area as well as on the surface of the hippocampus based on the usual pathology.

## 4. Discussion

The occurrence of MTLE with HS is rare. It originates from the limbic system in the mesial temporal lobe, particularly the hippocampus, amygdala, and parahippocampal gyrus, as well as their connections. HS is characterized pathologically by prominent neuronal loss and gliosis in the hippocampus and amygdala [13]. It is crucial to determine whether the seizure originates from the medial, lateral, or multifocal epileptogenic points because the diagnosis is difficult and therapeutic methods for each condition differ. The curative effect of the surgical treatment is well-established because the attack resolution rate after medical therapy is only 8% per year, whereas that after surgical treatment is 58% in the case of intractable TLE [14]. In addition, it is necessary to localize the epileptic focus preoperatively in several cases to prevent unnecessary brain resection. After the detailed localization of an epileptogenic focus through cortical mapping, certain invasive examinations such as cortical mapping with subdural electrodes may be required before moving on to resection as a second-stage surgery. However, it would be remarkable if a noninvasive examination could guide the diagnosis and localization of an epileptogenic focus.

If HS can be diagnosed preoperatively using noninvasive modalities such as MRI, a temporal lobectomy can be performed in a single stage [15–17]. Jack et al. (Mayo Clinic, 1990) noninvasively showed hippocampal atrophy and sclerotic change in an MRI coronal section; thereafter, assessing etiology, clinical course characteristics, EEG data, convalescence, surgical outcomes, and prognosis has become easy [18]. In 1993, Wieser indicated that MTLE is equivalent to an epileptic syndrome or as a disease unit [19]. Although MTLE is intractable, even with treatment using multiple anticonvulsant agents, it can be well-distinguished from their epileptic syndromes such as lateral TLE since they show surgical outcomes. Therefore, differential diagnosis is very important.

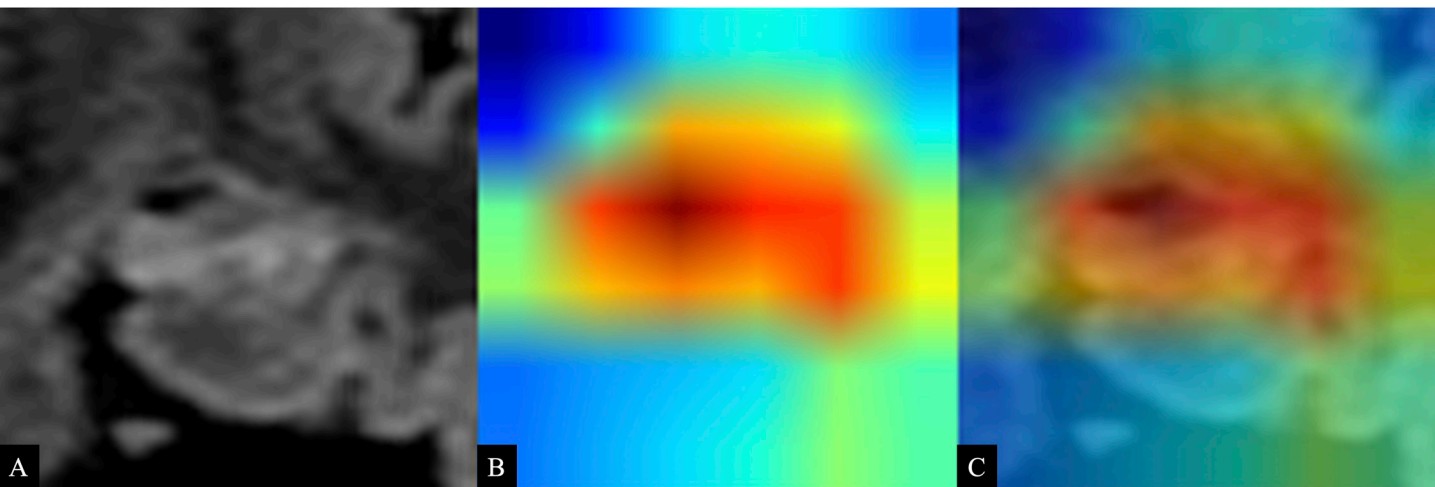

**Fig 5. Visualization of the area of interest by deep learning.** FLAIR image of a patient with MTLE with the left hippocampus as the epileptogenic area showing abnormal hyperintensity and atrophy (A). Visualization with color mapping using Grad-CAM technique (B) and fusion (C); areas of interest and "hot areas" crowded together surrounding the hippocampus and parahippocampal area.

The epidemiological cause of MTLE is HS, accounting for approximately two-thirds of all cases. Other causes include neoplastic lesions (such as ganglioglioma, glioma, and dysembryoplastic neuroepithelial tumor) and vascular lesions (such as arteriovenous malformations, cavernous angiomas or venous angiomas, and cortical dysplasia) [20]. Furthermore, MTLE associated with HS predominantly occurs among individuals aged is 4–16 years. MTLE due to HS generally occurs earlier than MTLE due to other causes [21].

MTLE is often associated with anamneses such as febrile convulsion (particularly complicated), brain hypoxia, an infectious disease, and/or head injury [21].

Hippocampal dysfunction initiates epileptogenic seizures and impairs cognition in patients with MTLE [22].

Recently, the use of AI in the medical care field has increased at a tremendous pace. For example, a patient was diagnosed with leukemia using the AI program within only 10 minutes after learning from more than 20 million cancer research articles, which proved lifesaving (Watson, IBM co.).

Several studies have presented a machine learning-based method for identifying the epileptic seizure onset zone [23–29]. The reason for the improvement is technical skillfulness and precise evaluation of the epileptogenic zone in the brain alongside functional mapping.

Recently, many scholars have been engaged in hippocampus segmentation algorithms. A number of methods have been proposed, including cluster methods [30], using map and feature embedding models or iterative local linear mapping models [31], atlas-based and label fusion methods [32], statistical machine learning techniques combined with KNN and SVM models [33], probabilistic modeling frame methods [34], nonlinear image registration algorithms [35], energy minimization models [36], using local Gaussian distribution fitting energy with level set function and local mean and variance as variables [37,38], and bias-corrected distance regularized level set method with MR image contrast enhancement [39].

Based on the aforementioned example, AI is suitable for processing a large amount of medical data (such as diagnostic imaging data), comparing case data with previous research data, and organizing data. Moreover, AI is a developing technology in the medical field. Previous data on rare diseases such as neurosurgical diseases are limited; hence, the accuracy may not be reliable because AI diagnosis requires a large amount of data. Here, we demonstrated an example of a diagnostic imaging technique for epilepsy.

As shown in Fig 5, AI focuses on the CA1 and CA2 regions of the hippocampus. In hippocampal sclerosis, pathological findings include not only neuronal atrophy but also neuronal loss and gliosis. Although not beyond the realm of conjecture, it is possible that AI can identify tiny differences in MRI signal values that cannot be corroborated by the physician's naked eye. We believe that the results of this study are helpful to specialists as well as non-specialists in community medicine for aiding in the diagnosis of epilepsy.

## 5. Limitations

Recently, deep learning technology has advanced rapidly; however, its ability to simultaneously diagnose several cases is suboptimal. In addition, the possibility of misdiagnosis because of bugs or malfunctioning of the software is inevitable. AI in medical care remains on the periphery of diagnostic science. Further research is required to learn and push the boundaries of the field.

## 6. Conclusion

We present a highly accurate deep learning-based AI program that can diagnose MTLE with the hippocampus as the epileptogenic area better than some board-certified neurosurgeons.

Almost all AI programs that are big hit worldwide are not all-purpose and perform limited functions such as extract and reproduce human specific data on the computer. However, AI is a powerful tool and can sometimes overcome the biases shown by humans. In addition, AI has the advantage of generating stable results. In the medical field, doctors have harsh working environments, which include overwork due to lack of personnel, frequent calls at night, and insufficient holidays; this may account for false judgments sometimes. On the contrary, AI can maintain a certain level of output accuracy and be a reliable assistant to doctors.

## Author Contributions

**Conceptualization:** Kyoya Sakashita, Yukinori Akiyama.

**Data curation:** Kyoya Sakashita.

**Formal analysis:** Yukinori Akiyama.

**Investigation:** Kyoya Sakashita, Tsukasa Hirano, Ayaka Sasagawa, Masayasu Arihara, Tomoyoshi Kuribara, Satoko Ochi, Rei Enatsu, Takeshi Mikami.

**Methodology:** Kyoya Sakashita, Yukinori Akiyama.

**Supervision:** Yukinori Akiyama, Nobuhiro Mikuni.

**Writing – original draft:** Kyoya Sakashita, Yukinori Akiyama.

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
