## [Decision Letter · Decision Letter 0]

24 May 2022

PONE-D-22-00437Deep Learning for the Diagnosis of Mesial Temporal Lobe EpilepsyPLOS ONE

Dear Dr. Sakashita,

Thank you for submitting your manuscript to PLOS ONE. After careful consideration, we feel that it has merit but does not fully meet PLOS ONE’s publication criteria as it currently stands. Therefore, we invite you to submit a revised version of the manuscript that addresses the points raised during the review process.

We look forward to receiving your revised manuscript.

Kind regards,

Yuvaraj Rajamanickam, Ph.D

Academic Editor

PLOS ONE

Journal Requirements:

Reviewers' comments:

Reviewer's Responses to Questions

**Comments to the Author**

1. Is the manuscript technically sound, and do the data support the conclusions?

Reviewer #1: No

Reviewer #2: Partly

2. Has the statistical analysis been performed appropriately and rigorously? 

Reviewer #1: No

Reviewer #2: Yes

3. Have the authors made all data underlying the findings in their manuscript fully available?

Reviewer #1: No

Reviewer #2: Yes

4. Is the manuscript presented in an intelligible fashion and written in standard English?

Reviewer #1: Yes

Reviewer #2: Yes

5. Review Comments to the Author

Reviewer #1: Firstly, I take this opportunity to congratulate the authors on their successful submission of their paper for publication.

Initially, I felt the paper can be recommended for publication with reasonable modifications. However, once I saw Fig. 3, I can no longer recommend the paper since there is something fundamentally wrong while training the deep learning model. The validation accuracy does not change even after training the model. This infers either the validation accuracy depicted in Fig. 3 is erroneous or the VGG16 model can detect MTLE even without re-training the model. Kindly rectify this issue since this is the main result of the paper.

Comments:

• The patient flow chart is well described. However, the number of patients in the training/validation set needs to be specified separately. Also, please indicate the number of images/augmented images from each patient. As of now, the data information in lines 114, 125, 157, 161, etc is confusing.

• Please specify how the training and validation data was selected. Was the dataset split at the patient-level or the image-level. The images from the same patient should not be included in both the training and validation set.

• I also recommend performing a cross-validation to verify the robustness of the methods prescribed in the manuscript.

• I am also concerned why 20 patients were specifically chosen to compare between the AI system and the clinicians. Ideally, the comparison should be performed for all the 46 MTLE patients. Please do not use a ‘convenient’ sample for comparisons.

• I also have concerns regarding the statistical testing:

o T-tests are performed if the data distribution is ‘normal’. I am not sure if the data points fall into a normal distribution as the sample size is too low. Please perform a ‘normality’ testing before applying a t-test. Alternatively, you need to apply Man-Whitney U test or Wilcoxon rank sum test.

o Kindly report the effect sizes along with p-values.

o In Table 3, for the first 10 patients, the ground truth is ‘1’ and for 11-20 patients the ground truth is ‘0’. Therefore, while performing the test, you will end up with two p-values (one for 1-10 patients and one for 11-20). Since you have reported a single p-value please specify how this was performed.

• Data augmentation is widely used while training the deep learning models. You should not apply them while computing the validation results as this will skew the accuracy since a single data point (patient) is considered as multiple observations. The validation/test results should be reported considering each patient as a single data point.

• Finally, if possible try to explain why the AI system was capable of diagnosing better than clinicians using Fig. 4. In other words, what did the AI system detect in the images that were missed by the clinicians.

Good luck.

Reviewer #2: 1. The authors failed to compare their segmentation results with hippocampal segmentation papers in the literature

2. The authors didn’t validate their segmentation accuracy through the deep learning cross validation methods

3. Authors didn’t explain about the architecture of deep learning in the manuscript

4. Authors didn’t compare their results with other segmentation algorithms or standard tool box

5. Authors didn’t explain the challenges of segmentation of hippocampus through deep learning algorithm

6. Authors gas to validate their results through other metrics like sensitivity, specificity, f1 score etc to verify the performance of algorithm

7. Authors has to explain the optimization of parameters of deep learning model

8. How MRI out performs the less cost EEG data in epilepsy diagnosis

9. Why T2 and FLAIR images are used instead of T1 weighted images?

10. Why there is huge difference between diagnostic accuracy of AI and neurosurgeon in table 3? Whether results can be validated with multiple surgeons results?

6. PLOS authors have the option to publish the peer review history of their article (what does this mean?). If published, this will include your full peer review and any attached files.

Reviewer #1: No

Reviewer #2: **Yes: **Jac Fredo Agastinose Ronickom

---

## [Author Response · Author response to Decision Letter 0]

1 Aug 2022

Reviewer #1: 

• Initially, I felt the paper can be recommended for publication with reasonable modifications. However, once I saw Fig. 3, I can no longer recommend the paper since there is something fundamentally wrong while training the deep learning model. The validation accuracy does not change even after training the model. This infers either the validation accuracy depicted in Fig. 3 is erroneous or the VGG16 model can detect MTLE even without re-training the model. Kindly rectify this issue since this is the main result of the paper.

RESPONSE: 

Thank you for your valuable comments. We completely agree with reviewer #1. The following section has been amended in the manuscript (p.12, line 276-281): 

The relatively high accuracy from the beginning of the epoch may be due firstly to the fact that VGG16 was a very good model for MTLE differentiation, and secondly to the fact that the image size was small (96 × 96 pixel) and the accuracy was high. However, considering that diagnostic accuracy increased with each successive epoch, we believe that there was a learning effect. 

• The patient flow chart is well described. However, the number of patients in the training/validation set needs to be specified separately. Also, please indicate the number of images/augmented images from each patient. As of now, the data information in lines 114, 125, 157, 161, etc is confusing.

RESPONSE:

Thank you very much for your valuable suggestion. We have added a table (Table 1) and organized the data. (p.7, line 177)

• Please specify how the training and validation data was selected. Was the dataset split at the patient-level or the image-level. The images from the same patient should not be included in both the training and validation set.

RESPONSE:

Thank you for your valuable question. We have now described how we split the training and validation datasets.

• I also recommend performing a cross-validation to verify the robustness of the methods prescribed in the manuscript.

RESPONSE:

Thank you for your valuable comment. We have now performed a cross-validation analysis, and the system showed a 90–93% accuracy for T2WI and an 89–95% accuracy for FLAIR.

We have included this information in line 287 (p.12) of the text.

• I am also concerned why 20 patients were specifically chosen to compare between the AI system and the clinicians. Ideally, the comparison should be performed for all the 46 MTLE patients. Please do not use a ‘convenient’ sample for comparisons.

RESPONSE:

Thank you for your comment. We have amended the text regarding this issue as follows (p.10, line 235):

Retrospectively, 46 patients were diagnosed with MTLE at our institution. However, we excluded cases with no imaging, cases in which surgery was performed, and cases for which no ECoG recording with subdural electrodes existed. Finally, T2WI was validated in 28 cases and FLAIR in 34 cases. 

And as follows (p.13, line 316): 

The diagnostic accuracy for MTLE by AI was higher than that by physicians for both T2WI and FLAIR (T2, p = 0.0034; effect size 0.3917, FLAIR, p = 0.0006; effect size, 0.4147). 　 

However, the sensitivity, specificity, and F-value of AI for T2WI were 0.5556, 0.9362, and 0.5582, respectively, and those of physicians for T2WI were 0.5000, 0.6754, and 0.4576, respectively. Similarly, in FLAIR, the sensitivity, specificity, and F-value for the diagnosis by AI were 0.4, 1.0000, and 0.5714, respectively, whereas those by physicians were 0.6833, 0.6181, and 0.5256, respectively. Although the F-value for AI was high, the sensitivity was low. 

• I also have concerns regarding the statistical testing:

o T-tests are performed if the data distribution is ‘normal’. I am not sure if the data points fall into a normal distribution as the sample size is too low. Please perform a ‘normality’ testing before applying a t-test. Alternatively, you need to apply Man-Whitney U test or Wilcoxon rank sum test.

o Kindly report the effect sizes along with p-values.

o In Table 3, for the first 10 patients, the ground truth is ‘1’ and for 11-20 patients the ground truth is ‘0’. Therefore, while performing the test, you will end up with two p-values (one for 1-10 patients and one for 11-20). Since you have reported a single p-value please specify how this was performed.

RESPONSE:

Thank you for your comments. The Mann-Whitney U test was used to validate the data. (p.11, line 243)

The following text has also been amended in the manuscript (p.12, line 288):

Those judged to have a 50% or greater likelihood of hippocampal sclerosis, according to the AI diagnosis, were treated as correct answers. Those that were not hippocampal sclerosis were treated as correct if the likelihood of hippocampal sclerosis was judged to be less than 50%. 

• Data augmentation is widely used while training the deep learning models. You should not apply them while computing the validation results as this will skew the accuracy since a single data point (patient) is considered as multiple observations. The validation/test results should be reported considering each patient as a single data point.

RESPONSE:

Thank you for your salient comments. The following text has been amended (p.11, line 271):

When the diagnostic accuracy was measured using only the original images as the test data, the diagnostic accuracy using T2WI data was 98%, and that using FLAIR data was 95%. However, this result is considered to be insufficiently evaluated because of the small number of original images. 

• Finally, if possible try to explain why the AI system was capable of diagnosing better than clinicians using Fig. 4. In other words, what did the AI system detect in the images that were missed by the clinicians

RESPONSE:

Thank you for your comment. The following text has been amended in response (p.16, line 391):

As shown in Fig. 5, AI focuses on the CA1 and CA2 regions of the hippocampus. In hippocampal sclerosis, pathological findings include not only neuronal atrophy but also neuronal loss and gliosis. Although not beyond the realm of conjecture, it is possible that AI can identify tiny differences in MRI signal values that cannot be corroborated by the physician's naked eye. 

Reviewer #2: 

1. The authors failed to compare their segmentation results with hippocampal segmentation papers in the literature

RESPONSE:

Thank you for your valuable comment. The following sentences have been added to the text (p.15, line 376): 

Recently, many scholars have engaged in hippocampus segmentation algorithms. Several methods have been proposed, including cluster methods (Pang S), using map and feature embedding models or iterative local linear mapping models (Shumao P), atlas-based and label fusion methods (Carmichael O T), statistical machine learning techniques combined with KNN and SVM models (Hao Y), probabilistic modeling frame methods (Platero C), nonlinear image registration algorithms (Zhu H), energy minimization models (Lijn F V D), using local Gaussian distribution fitting energy with a level set function and local mean and variance as variables (Xiaoliang J, Wang L), and bias-corrected distance regularized level set method with MR image contrast enhancement (Selma T).

2. The authors didn’t validate their segmentation accuracy through the deep learning cross validation methods

RESPONSE:

Thank you for your valuable comment. We performed a cross-validation analysis, and the system showed 90–93% accuracy for T2WI and 89–95% accuracy for FLAIR data.

We have included this information in line 288 (p.12) of the text.

3. Authors didn’t explain about the architecture of deep learning in the manuscript

RESPONSE:

Thank you for your appropriate comments. We have included this figure.

Details of the neural networks of the model are shown in Fig. 2. (p7, line 189)

4. Authors didn’t compare their results with other segmentation algorithms or standard tool box

RESPONSE:

Thank you for your important remarks. We have included the information in the text, on page 11, starting at line 259, as below: 

Haar-like feature was selected because it is more accurate than LBP when the hyperparameters are changed to optimize the detection rate and reduce false positives. FLAIR had a 89% detection rate and 0.25 false positives per picture. While checking the detection rate and false positive rate, we examined the effect of changing mainly the following parameters as optimization factors: scaleFactor, minNeighbors, minSize, and maxSize. For the T2WI classifier, scaleFactor was set to 1.06, minNeighbors to 3, minSize to (60,60), and maxSize to (130,130). Similarly, for FLAIR, we set the scaleFactor to 1.1202, minNeighbors to 3, minSize to (60,60), and maxSize to (130,130).

5. Authors didn’t explain the challenges of segmentation of hippocampus through deep learning algorithm

RESPONSE:

Thank you for your important remarks. We have included this information in line 156 (p.5) in the text, as follows:

OpenCV was used for hippocampal detection. MRI images were obtained of 113 T2WI and 148 FLAIR cases with a diagnosis of epilepsy from January 2016 to December 2019. On both the left and right sides, 226 T2WI and 296 FLAIR images with a resolution of 96 × 96 pixels were used for hippocampal images, and 327 T2WI and 367 FLAIR images with a resolution of 96 × 96 pixels were used for non-hippocampal images after amplification using augmentation. Haar-like features and local binary patterns (LBPs) were used as features. Hyperparameters of the created cascade classifier, mainly scaleFactor, minNeighbors, and minSize, were adjusted for optimization. 

6. Authors gas to validate their results through other metrics like sensitivity, specificity, f1 score etc to verify the performance of algorithm

RESPONSE:

Thank you for your valuable comment. The following sentences have been added to the text:

(p.12, line 298):

Diagnoses by AI were significantly more accurate than those by board-certified neurosurgeons based on T2WI images (p = 0.0001, effect size 0.6152); however, the difference in diagnostic accuracies of AI and neurosurgeons based on FLAIR was not statistically significant (p = 0.0520, effect size 0.3072) (Table 4).

(p.12, line 303):

However, the sensitivity, specificity, and F-value of the AI for T2WI were 0.8000, 0.9667, and 0.8421, respectively, and those of physicians for T2WI were 0.5000, 0.5667, and 0.5172, respectively. Similarly, for FLAIR, the sensitivity, specificity, and F-value for the diagnosis by AI were 0.3077, 1.0000, and 0.4706, respectively, whereas those by physicians were 0.5833, 0.5833, and 0.5833, respectively. Although the sensitivity, specificity, and F-value of AI for T2WI were high, the sensitivity for FLAIR was low.

(p. 13, line 317):

The diagnostic accuracy for MTLE by AI was higher than that by physicians for both T2WI and FLAIR (T2, p = 0.0034; effect size 0.3917, FLAIR, p = 0.0006; effect size, 0.4147). 　 

However, the sensitivity, specificity, and F-value of AI for T2WI were 0.5556, 0.9362, and 0.5582, respectively, and those of physicians for T2WI were 0.5000, 0.6754, and 0.4576, respectively. Similarly, in FLAIR, the sensitivity, specificity, and F-value for the diagnosis by AI were 0.4, 1.0000, and 0.5714, respectively, whereas those by physicians were 0.6833, 0.6181, and 0.5256, respectively. Although the F-value for AI was high, the sensitivity was low.

7. Authors has to explain the optimization of parameters of deep learning model

RESPONSE:

Thank you for your important remarks. 

We have now added the following sentence in the text (p.7, line 190):

We used SGD as an optimizer for the deep learning model.

8. How MRI out performs the less cost EEG data in epilepsy diagnosis

RESPONSE:

Thank you for your very important question.

We believe that AI will be useful for screening. We do not mean, of course, that EEG data are unnecessary, nor do we mean that MRI is superior to EEG. We hope that both MRI and EEG data will be used for more accurate diagnosis of MTLE in the future.

9. Why T2 and FLAIR images are used instead of T1 weighted images?

RESPONSE:

Thank you for your very important question.

In general clinical practice, physicians often diagnose hippocampal sclerosis using T2 rather than T1. This is thought to be because atrophy and sclerosis are more coordinated in T2. Therefore, T2WI and FLAIR were used in this study.

10. Why there is huge difference between diagnostic accuracy of AI and neurosurgeon in table 3? Whether results can be validated with multiple surgeons results?

RESPONSE:

Thank you for your very important question. We have addressed this in the following lines (p.16, line 391):

As shown in Fig. 5, AI focuses on the CA1 and CA2 regions of the hippocampus. In hippocampal sclerosis, pathological findings include not only neuronal atrophy but also neuronal loss and gliosis. Although not beyond the realm of conjecture, it is possible that AI can identify tiny differences in MRI signal values that cannot be corroborated by the physician's naked eye.

---

## [Decision Letter · Decision Letter 1]

14 Sep 2022

PONE-D-22-00437R1Deep Learning for the Diagnosis of Mesial Temporal Lobe EpilepsyPLOS ONE

Dear Dr. Sakashita,

Thank you for submitting your manuscript to PLOS ONE. After careful consideration, we feel that it has merit but does not fully meet PLOS ONE’s publication criteria as it currently stands. Therefore, we invite you to submit a revised version of the manuscript that addresses the points raised during the review process.

We look forward to receiving your revised manuscript.

Kind regards,

Yuvaraj Rajamanickam, Ph.D

Academic Editor

PLOS ONE

Reviewers' comments:

Reviewer's Responses to Questions

**Comments to the Author**

1. If the authors have adequately addressed your comments raised in a previous round of review and you feel that this manuscript is now acceptable for publication, you may indicate that here to bypass the “Comments to the Author” section, enter your conflict of interest statement in the “Confidential to Editor” section, and submit your "Accept" recommendation.

Reviewer #1: (No Response)

Reviewer #2: All comments have been addressed

2. Is the manuscript technically sound, and do the data support the conclusions?

Reviewer #1: Partly

Reviewer #2: Yes

3. Has the statistical analysis been performed appropriately and rigorously? 

Reviewer #1: No

Reviewer #2: Yes

4. Have the authors made all data underlying the findings in their manuscript fully available?

Reviewer #1: Yes

Reviewer #2: Yes

5. Is the manuscript presented in an intelligible fashion and written in standard English?

Reviewer #1: No

Reviewer #2: Yes

6. Review Comments to the Author

Reviewer #1: Firstly, I take this opportunity to congratulate the authors on their successful re-submission of their paper for publication.

Comments:

1. “Initially, I felt the paper can be recommended for publication with reasonable modifications. However, once I saw Fig. 3, I can no longer recommend the paper since there is something fundamentally wrong while training the deep learning model. The validation accuracy does not change even after training the model. This infers either the validation accuracy depicted in Fig. 3 is erroneous or the VGG16 model can detect MTLE even without re-training the model. Kindly rectify this issue since this is the main result of the paper.”

The relatively high accuracy from the beginning of the epoch may be due firstly to the fact that VGG16 was a very good model for MTLE differentiation, and secondly to the fact that the image size was small (96 × 96 pixel) and the accuracy was high. However, considering that diagnostic accuracy increased with each successive epoch, we believe that there was a learning effect.

Please elaborate the statement that the “diagnostic accuracy is increased in a successive epoch”. As observed from the Figure 4, the validation accuracy is decreasing or remaining the same throughout the training. The training accuracy is expected to increase and saturate overtime. The validation accuracy typically represents the results on an unknown test set. Therefore, if the validation accuracy is high without any training, it means that the model without any training is can perform classification task. Ideally, the validation accuracy should increase with training and later saturate or decrease due to overfitting.

2. The patient flow chart is well described. However, the number of patients in the training/validation set needs to be specified separately. Also, please indicate the number of images/augmented images from each patient. As of now, the data information in lines 114, 125, 157, 161, etc is confusing.

Thank you very much for your valuable suggestion. We have added a table (Table 1) and organized the data. (p.7, line 177)

Thank you for adding a Tables 2 & 3 regarding patient information. However, please combine these two tables since I found that most of the patients were overlapping between the two analyses.

The Table 1 is interesting. Please include details regarding the ‘validation’ set. Also, I find it consuming why data augmentation was performed on the test set. Data augmentation is performed on the training set to increase the number of observations and to make the system robust to noise. If it was applied on the test set, you are typically skewing the performance metrics. I understood you have presented both the results, but the results based on data augmentation is skewed.

3. I also recommend performing a cross-validation to verify the robustness of the methods prescribed in the manuscript.

Thank you for your valuable comment. We have now performed a cross-validation analysis, and the system showed a 90–93% accuracy for T2WI and an 89–95% accuracy for FLAIR. We have included this information in line 287 (p.12) of the text.

Please specify the details of cross-validation: How many folds? How was the hyper parameters optimized? etc.

4. I am also concerned why 20 patients were specifically chosen to compare between the AI system and the clinicians. Ideally, the comparison should be performed for all the 46 MTLE patients. Please do not use a ‘convenient’ sample for comparisons.

I also have concerns regarding the statistical testing:

T-tests are performed if the data distribution is ‘normal’. I am not sure if the data points fall into a normal distribution as the sample size is too low. Please perform a ‘normality’ testing before applying a t-test. Alternatively, you need to apply Man-Whitney U test or Wilcoxon rank sum test.

Kindly report the effect sizes along with p-values.

In Table 3, for the first 10 patients, the ground truth is ‘1’ and for 11-20 patients the ground truth is ‘0’. Therefore, while performing the test, you will end up with two p-values (one for 1-10 patients and one for 11-20). Since you have reported a single p-value please specify how this was performed.

Those judged to have a 50% or greater likelihood of hippocampal sclerosis, according to the AI diagnosis, were treated as correct answers. Those that were not hippocampal sclerosis were treated as correct if the likelihood of hippocampal sclerosis was judged to be less than 50%.

The reply to this question is incomplete. Please specify whether you have performed normality testing of the feature distribution to choose Mann-Whitney U test. Also, specify the type of d-values presented.

Now, the modification on p 12, line 288 have given more concerns regarding the statistical testing. Mann-Whitney U test is performed to assess if two distribution as significantly different. However, since the authors claim that the prediction values were converted to binary as correct or wrong, then this statistical testing cannot be applied.

Reviewer #2: Authors has addressed all the comments of the reviewer. I recommend the manuscript for publication.

7. PLOS authors have the option to publish the peer review history of their article (what does this mean?). If published, this will include your full peer review and any attached files.

Reviewer #1: No

Reviewer #2: No

---

## [Author Response · Author response to Decision Letter 1]

26 Oct 2022

Thank you for your thoughtful and constructive feedback regarding our manuscript titled “Deep Learning for the Diagnosis of Mesial Temporal Lobe Epilepsy.”

We also appreciate the time and effort you and each of the reviewers have dedicated to providing suggestions, which have strengthened our paper. Thus, it is with great pleasure that we resubmit our article for further consideration. We have incorporated changes that reflect your suggestions. We hope that our edits and responses satisfactorily address all of the noted issues and concerns.

Reviewer #1: 

• Please elaborate the statement that the “diagnostic accuracy is increased in a successive epoch”. As observed from the Figure 4, the validation accuracy is decreasing or remaining the same throughout the training. The training accuracy is expected to increase and saturate overtime. The validation accuracy typically represents the results on an unknown test set. Therefore, if the validation accuracy is high without any training, it means that the model without any training is can perform classification task. Ideally, the validation accuracy should increase with training and later saturate or decrease due to overfitting.

RESPONSE: 

Thank you for your valuable comment. Although it is difficult to understand which epoch the horizontal axis is referring to in the graph as the number of iterations increases within 1 epoch, accuracy increases to about 0.8 within the 1st epoch. In this learning, the accuracy almost reaches a plateau at 2 epochs, making this difficult to reflect it in the graph.

• Thank you for adding a Tables 2 & 3 regarding patient information. However, please combine these two tables since I found that most of the patients were overlapping between the two analyses.

The Table 1 is interesting. Please include details regarding the ‘validation’ set. Also, I find it consuming why data augmentation was performed on the test set. Data augmentation is performed on the training set to increase the number of observations and to make the system robust to noise. If it was applied on the test set, you are typically skewing the performance metrics. I understood you have presented both the results, but the results based on data augmentation is skewed.

RESPONSE:

Thank you for your valuable suggestion. We have compiled the data previously presented in Tables 2 and 3 in the new Table 3. The data in Table 1 has been revised, with the amplified validation data deleted. We have added the details regarding the validation data in the new Table 2. Cross-validation was performed using three of these cases as validation data. 

• Please specify the details of cross-validation: How many folds? How was the hyper parameters optimized? etc.

RESPONSE:

For MTLE with the hippocampus as the epileptogenic area, three out of nine T2 cases and 10 FLAIR cases were selected as validation data. Verification was performed five times by replacing the data. We obtained results of 90–93% accuracy for T2WI and 89–95% accuracy for FLAIR.

• The reply to this question is incomplete. Please specify whether you have performed normality testing of the feature distribution to choose Mann-Whitney U test. Also, specify the type of d-values presented.

Now, the modification on p 12, line 288 have given more concerns regarding the statistical testing. Mann-Whitney U test is performed to assess if two distribution as significantly different. However, since the authors claim that the prediction values were converted to binary as correct or wrong, then this statistical testing cannot be applied.

RESPONSE:

Thank you for your valuable comment. The Shapiro–Wilk test showed that the data did not follow a normal distribution (p<0.000); therefore, Mann–Whitney U test was performed.

Thank you again for giving us the opportunity to strengthen our manuscript with your valuable comments and suggestions. We have worked hard to incorporate your feedback, and hope that our revised manuscript is suitable for publication.

Yours sincerely,

---

## [Decision Letter · Decision Letter 2]

22 Nov 2022

PONE-D-22-00437R2Deep Learning for the Diagnosis of Mesial Temporal Lobe EpilepsyPLOS ONE

Dear Dr. Sakashita,

Thank you for submitting your manuscript to PLOS ONE. After careful consideration, we feel that it has merit but does not fully meet PLOS ONE’s publication criteria as it currently stands. Therefore, we invite you to submit a revised version of the manuscript that addresses the points raised during the review process.

We look forward to receiving your revised manuscript.

Kind regards,

Yuvaraj Rajamanickam, Ph.D

Academic Editor

PLOS ONE

Journal Requirements:

Reviewers' comments:

Reviewer's Responses to Questions

**Comments to the Author**

1. If the authors have adequately addressed your comments raised in a previous round of review and you feel that this manuscript is now acceptable for publication, you may indicate that here to bypass the “Comments to the Author” section, enter your conflict of interest statement in the “Confidential to Editor” section, and submit your "Accept" recommendation.

Reviewer #1: (No Response)

Reviewer #2: All comments have been addressed

2. Is the manuscript technically sound, and do the data support the conclusions?

Reviewer #1: Partly

Reviewer #2: Yes

3. Has the statistical analysis been performed appropriately and rigorously? 

Reviewer #1: No

Reviewer #2: Yes

4. Have the authors made all data underlying the findings in their manuscript fully available?

Reviewer #1: No

Reviewer #2: Yes

5. Is the manuscript presented in an intelligible fashion and written in standard English?

Reviewer #1: No

Reviewer #2: Yes

6. Review Comments to the Author

Reviewer #1: Firstly, I take this opportunity to congratulate the authors on their successful re-submission of their paper for publication.

Comments:

RESPONSE:

Thank you for your valuable comment. Although it is difficult to understand which

epoch the horizontal axis is referring to in the graph as the number of iterations

increases within 1 epoch, accuracy increases to about 0.8 within the 1st epoch. In this

learning, the accuracy almost reaches a plateau at 2 epochs, making this difficult to

reflect it in the graph.

Question: This is exactly where I am confused. Within 2 epochs, the model saturated and the validation accuracy started decreasing. This just shows that the VGG16 model can detect MTLE even without re-training the model. This just shows that the entire training process is redundant in this study.

RESPONSE:

Thank you for your valuable suggestion. We have compiled the data previously

presented in Tables 2 and 3 in the new Table 3. The data in Table 1 has been revised,

with the amplified validation data deleted. We have added the details regarding the

validation data in the new Table 2. Cross-validation was performed using three of

these cases as validation data.

RESPONSE:

For MTLE with the hippocampus as the epileptogenic area, three out of nine T2 cases

and 10 FLAIR cases were selected as validation data. Verification was performed five

times by replacing the data. We obtained results of 90–93% accuracy for T2WI and

89–95% accuracy for FLAIR.

Question: Please elaborate on this procedure of replacing the data. Since this cross-validation results are the key results of the paper it needs to be well detailed.

RESPONSE:

Thank you for your valuable comment. The Shapiro–Wilk test showed that the data did

not follow a normal distribution (p<0.000); therefore, Mann–Whitney U test was

performed.

Question: The answer to this question is incomplete. “The reply to this question is incomplete. Please specify whether you have performed normality testing of the feature distribution to choose Mann-Whitney U test. Also, specify the type of d-values presented.

Now, the modification on p 12, line 288 have given more concerns regarding the statistical testing. Mann-Whitney U test is performed to assess if two distribution as significantly different. However, since the authors claim that the prediction values were converted to binary as correct or wrong, then this statistical testing cannot be applied.”

Reviewer #2: Reviewer has given his decision to accept the paper in the previous revision. Editor can take the final decision regarding the publication of the draft in the journal.

7. PLOS authors have the option to publish the peer review history of their article (what does this mean?). If published, this will include your full peer review and any attached files.

Reviewer #1: No

Reviewer #2: No

---

## [Author Response · Author response to Decision Letter 2]

29 Dec 2022

Reviewer #1: 

Question: This is exactly where I am confused. Within 2 epochs, the model saturated and the validation accuracy started decreasing. This just shows that the VGG16 model can detect MTLE even without re-training the model. This just shows that the entire training process is redundant in this study.

RESPONSE: 

Thank you for pointing this out. We apologize for not being able to give a clear answer. To check whether the VGG16 model can detect MTLE, we performed 20 diagnoses that were also used in the paper with a 1-epoch learning model. The result was not limited to whether MTLE was true or false, and the probability of MTLE was between 0.10 and 0.41 for FLAIR and between 0.01 and 0.55 for T2, which were not diagnostically useful. Although it is not a direct answer, it shows that it is necessary to repeat the number of times of learning.

Question: Please elaborate on this procedure of replacing the data. Since this cross-validation results are the key results of the paper it needs to be well detailed.

RESPONSE:

A random number was used to assign which cases were to be trained and which were to be tested. (p.11, line 281)

Question: The answer to this question is incomplete. “The reply to this question is incomplete. Please specify whether you have performed normality testing of the feature distribution to choose Mann-Whitney U test. Also, specify the type of d-values presented.

Now, the modification on p 12, line 288 have given more concerns regarding the statistical testing. Mann-Whitney U test is performed to assess if two distribution as significantly different. However, since the authors claim that the prediction values were converted to binary as correct or wrong, then this statistical testing cannot be applied.”

RESPONSE:

Thank you for pointing this out. I apologize for my lack of knowledge about statistics. The Shapiro–Wilk test showed that the data did not follow a normal distribution (p<0.000); therefore, Mann-Whitney U test was performed. (p.13, line 322)

A cross tabulation was performed to compare the number of correct and incorrect numbers for AI and Dr. Results have shown that AI is superior in diagnosis, and we believe that we have demonstrated the effectiveness of AI. (p.13, line 325)

---

## [Decision Letter · Decision Letter 3]

8 Feb 2023

Deep Learning for the Diagnosis of Mesial Temporal Lobe Epilepsy

PONE-D-22-00437R3

Dear Dr. Sakashita,

We’re pleased to inform you that your manuscript has been judged scientifically suitable for publication and will be formally accepted for publication once it meets all outstanding technical requirements.

Kind regards,

Yuvaraj Rajamanickam, Ph.D

Academic Editor

PLOS ONE

Additional Editor Comments (optional):

I understand that there are mix opinios on this paper. Overall, the data interpretation and results in this manuscript are useful and interesting to the PLOSONE readers. The conclusions are well suppoted with the data.

Reviewers' comments:

Reviewer's Responses to Questions

**Comments to the Author**

1. If the authors have adequately addressed your comments raised in a previous round of review and you feel that this manuscript is now acceptable for publication, you may indicate that here to bypass the “Comments to the Author” section, enter your conflict of interest statement in the “Confidential to Editor” section, and submit your "Accept" recommendation.

Reviewer #1: (No Response)

2. Is the manuscript technically sound, and do the data support the conclusions?

Reviewer #1: No

3. Has the statistical analysis been performed appropriately and rigorously? 

Reviewer #1: No

4. Have the authors made all data underlying the findings in their manuscript fully available?

Reviewer #1: No

5. Is the manuscript presented in an intelligible fashion and written in standard English?

Reviewer #1: Yes

6. Review Comments to the Author

Reviewer #1: Dear Authors,

I still find the answers to my reviews incomplete and therefore, I cannot recommend the manuscript for publication.

1. I did not understand this answer: "The result was not limited to whether MTLE was true or false, and the probability of MTLE was between 0.10 and 0.41 for FLAIR and between 0.01 and 0.55 for T2, which were not diagnostically useful". My question was since from the accuracy figure (Fig. 4) the VGG16 model performs better on the validation data without any training. Why do you then add additional training and make the performance inferior?

2. Cross-validation need to be systematically performed in this study.

3. I still did not understand how the statistical testing was performed. I find multiple p values all over the paper. Also, the specific type of d-value used in the manuscript needs to be defined.

7. PLOS authors have the option to publish the peer review history of their article (what does this mean?). If published, this will include your full peer review and any attached files.

Reviewer #1: No

<quillbot-extension-portal></quillbot-extension-portal>

---

## [Editor Report · Acceptance letter]

10 Feb 2023

PONE-D-22-00437R3 

Deep Learning for the Diagnosis of Mesial Temporal Lobe Epilepsy 

Dear Dr. Sakashita:

I'm pleased to inform you that your manuscript has been deemed suitable for publication in PLOS ONE. Congratulations! Your manuscript is now with our production department. 

Kind regards, 

on behalf of

Dr. Yuvaraj Rajamanickam 

Academic Editor

PLOS ONE